

# Watch-wearing as a marker of conscientiousness

David A. Ellis[1] and Rob Jenkins[2]

[1] Department of Psychology, Lancaster University, UK
[2] Department of Psychology, University of York, UK

## ABSTRACT

Several aspects of an individual's appearance have been shown to predict personality and related behaviour. While some of these cues are grounded in biology (e.g., the human face), other aspects of a person's appearance can be actively controlled (e.g., clothing). In this paper, we consider a common fashion accessory, the wristwatch. In an exploratory sample ($N > 100$) and a confirmatory sample ($N > 600$), we compared big-five personality traits between individuals who *do* or *do not* regularly wear a standard wristwatch. Significantly higher levels of conscientiousness were observed in participants who wore a watch. In a third study ($N = 85$), watch wearers arrived significantly earlier to appointments in comparison to controls. These results are discussed in relation to enclothed cognition and the rise of wearable technology including smartwatches.

## INTRODUCTION

Observers routinely make rapid inferences about personality based on aspects of personal appearance across a variety of contexts (*Wall et al., 2013*). Inferences are often based on information revealed through cues from the face, body, or voice. For example, aspects of personality extracted from brief snippets of novel voices are remarkably consistent between participants (*McAleer, Todorov & Belin, 2014*). People with broad faces are consistently rated as aggressive (*Carré & McCormick, 2008*). For some traits, there appears to be a strong biological basis that explains any behavioural correlate—testosterone affects facial appearance and aggression for example (*Verdonck et al., 1999*). However, a second related branch of research concerns other aspects of an individuals' appearance that can actively be controlled and a variety of specific inferential links have been observed between particular 'features' of clothing and components of character. Participants who wear glasses were rated as less extraverted and less open to experience (*Borkenau, 1991*; *Hellstrom & Tekle, 2006*) while the presence of tattoos are associated with lower levels of conscientious and higher levels of extraversion (*Swami, 2012*).

This line of research also raises the question of how reliable these inferences are in terms of predicting behaviour. The fact that these facets of appearance are chosen by the individual rather than being biologically endowed may suggest a weaker link between appearance and behaviour, but a growing body of research on the phenomenon of 'enclothed cognition', where changes in clothing can also affect behaviour, challenges this assumption.

Corresponding author
David A. Ellis,
d.a.ellis@lancaster.ac.uk

*Adam & Galinsky (2012)* recently demonstrated that wearing a lab coat described as a 'doctor's coat' increased sustained attention when compared to wearing a lab coat that was labeled as a 'painter's coat.' They argue that 'enclothed cognition' depends on both the symbolic meaning and the physical experience of wearing clothes. In addition, effects running in the opposite direction (from personality to appearance) may be more plausible for non-biological factors. An aggressive person for instance cannot choose to have a broader face, but he could choose to wear black clothes and make themselves appear more aggressive (*Vrij, 1997*). Here we focus on one particular clothing accessory, the wristwatch. Watches are an interesting case because they are designed to perform a very specific function—to tell the time. This specificity of function lends itself to experimentation because it suggests very targeted predictions about personality and behaviour.

Despite the rise in mobile devices with built-in clocks, the number of standard watch owners has remained static in recent years (*Hoffman, 2009*; *Mintel, 2010*). On the other hand, while many people continue to regularly wear a wristwatch, many chose to avoid them completely. Their prominence or absence in everyday life again makes them an ideal candidate when considering external markers of personality.

While research concerning the relationship between personality and an individual's outward appearance appears to be flourishing (e.g., *Hellstrom & Tekle, 2006*; *Gillath et al., 2012*; *Swami, 2012*), a number of limitations continue to affect this literature. First, there remains an over-reliance on university student samples. These samples may not be representative of the wider population (*Swami, 2012*). Secondly, previous research often fails to go beyond self-report (e.g., *Gillath et al., 2012*), with many papers failing to include an additional behavioural measure that may help explain or confirm differences observed in personality scores alone.

In order to overcome these limitations, and based on the premise that a core component of conscientiousness is good timekeeping, planning (*Back, Schmukle & Egloff, 2006*), and organisation (*Lee & Ashton, 2004*), we predicted that watch wearers would score consistently higher on a simple measure of conscientiousness in comparison to non-watch wearers. Accordingly, timekeeping can be operationalised as punctuality and if watch wearers really are more conscientious then they will, in turn, be more punctual in a real-life setting.

## STUDY 1

### Ethics statement

The University of Glasgow, College of Science & Engineering Ethics Committee approved all research (2013–4641). Participants were informed about procedures in detail and provided written informed consent.

### Method

#### Measures

We assessed personality using the Ten-Item Personality Inventory (TIPI). The TIPI was developed by *Gosling, Rentfrow & Swann (2003)* to meet the need for a very brief measure of the Big-Five personality dimensions (extraversion, agreeableness, conscientiousness,

**Table 1** Personality differences between watch and non-watch wearers in an exploratory sample.

Exploratory sample $N = 112$

| | $\alpha$ | Watch | | $t$ | $d$ |
| | | Yes $n = 53$ | No $n = 59$ | | |
| --- | --- | --- | --- | --- | --- |
| Extraversion | .48 | 4.53 (1.17) | 4.69 (1.30) | .67 | −.13 |
| Agreeableness | .13 | 4.49 (1.32) | 4.73 (.88) | 1.13 | −.22 |
| Conscientiousness | .63 | 5.35 (1.54) | 4.31 (1.24) | 3.94[*] | .75 |
| Emotional stability | .45 | 4.65 (1.31) | 4.57 (1.21) | .35 | .07 |
| Openness to experiences | .39 | 5.18 (1.26) | 5.46 (.98) | 1.31 | −.25 |

**Notes.**

[*] $p < .05$. Standard Deviations appear in parenthesis alongside means.

emotional stability and openness to experience). This measure was chosen due to its short nature, which allowed us to collect comparable data from both members of the public and students who had a limited amount of time to take part.

### Participants

One hundred and twelve participants were recruited and included members of the public attending The British Science Festival in 2010 and students studying psychology at Glasgow or Lincoln Universities in the United Kingdom (62.5% female) who were waiting to take part in experiments. Their ages ranged from 17 to 54.

### Procedure

Individuals approaching a psychology stand were asked if they wished to take part in a short study related to personality. If written consent was obtained, participants were required to fill out the TIPI. They were then asked whether or not they regularly wore a wristwatch. A regular watch wearer was defined as someone who wore a standard wristwatch, most of the time, for at least a year. Finally, all participants were thanked for their time and fully debriefed as to the true nature of the study.

### Results

As expected, participants who identified themselves as regular watch wearers rated themselves as significantly more conscientious when compared with controls (Table 1). We also observed that watch wearers scored lower in extraversion, agreeableness and openness, but higher on emotional stability. However, before conducting a further multivariate analysis, we next sought to replicate this finding in a larger confirmatory sample.

## STUDY 2

We attempted to replicate the results from Study 1 in a large online sample who, after completing the TIPI were asked:

'*Do you regularly wear a watch?*'

Participants were recruited via numerous email shots and twitter advertisements. They also provided information about their age, gender, location, working habits and mobile

phone ownership. In total, 638 participants took part (48.6% female). Modal age bands were 35–54 (36.4%) and 18–24 (30.5%); modal locations UK (60.8%), North America (13%). Regarding working habits, 49.7% confirmed that they worked a traditional Monday–Friday dayshift with the remainder working alternative hours (e.g., shifts, unemployed or students). Finally, 45.5% percent ($N = 290$) identified themselves as being regular watch wearers.

## Preliminary analysis

A preliminary analysis revealed no significant difference in the distribution of genders between the watch and non-watch groups ($X^2$ (1, $N = 632$) = 2.36, $p = .124$). While 97.48% of our sample owned a mobile phone, we also observed that there was no significant difference in this distribution of phone ownership between watch and non-watch wearers ($X^2(1, N = 635) = .803, p = .370$). Finally, there was no significant difference in the distribution of those who worked traditional or shift based work between watch and non-watch groups ($X^2(1, N = 637) = .680, p = .410$).

## Replication of study 1

An independent sample $t$-test again revealed significant differences in mean conscientiousness scores between watch and non-watch wearers (Table 2). Further $t$-tests revealed no other significant personality differences between watch and non-watch wearers across the other four factors of personality [$p$'s $> .05$]. As observed in Study 1, however, we again observed similar trends whereby watch wearers scored lower in extraversion and openness in comparison to controls.

## Regression model

In order to confirm that the personality differences reported above hold after controlling for additional factors, we entered age, gender and all five personality factors into a binary logistic model. This model confirms that wearing a watch remains a visible indicator for

**Table 2 Personality differences between watch and non-watch wearers in a confirmatory sample.**

Confirmatory sample $N = 638$

| | | Watch | | | |
| | | Yes | No | | |
| | $\alpha$[a] | $n = 290$ | $n = 348$ | $t$ | $d$ |
|---|---|---|---|---|---|
| Extraversion | .77 | 3.83 (1.57) | 3.90 (1.60) | .55 | −.04 |
| Agreeableness | .36 | 4.71 (1.20) | 4.64 (1.22) | .80 | .06 |
| Conscientiousness | .58 | 4.81 (1.39) | 4.56 (1.37) | 2.21[*] | .18 |
| Emotional stability | .66 | 4.53 (1.48) | 4.57 (1.46) | .33 | −.03 |
| Openness to experiences | .41 | 5.14 (1.15) | 5.32 (1.15) | 1.89 | −.01 |

Notes.
[*] $p < .05$. Standard Deviations appear in parenthesis alongside means.
[a] The TIPI was intentionally designed to produce low coefficient alphas, which are themselves misleading when calculated on scales with a reduced number of items (*Kline, 2000*; *Wood & Hampson, 2005*). Our reported values compare favorably to the internal measures of consistency observed by *Gosling, Rentfrow & Swann (2003)* during the scales initial development.

conscientiousness even after controlling for gender and age (Table 3). In other words, the odds of wearing a watch is significantly larger for a person who reports higher levels of conscientiousness (odds ratio = 1.147).

## Multivariate analysis

Personality is a multidimensional construct and effect sizes should also be considered in relation to the overall magnitude of differences observed between two groups. When groups differ along several variables at once, the overall between-group difference is not always accurately represented by *univariate* effect sizes in isolation. Therefore, *Del Giudice, Booth & Irwing (2012)* have argued that in order to aggregate differences across variables while also taking correlation patterns into account, it is necessary to computer a *multivariate* effect size. The Mahalanobis distance $D$ metric allows for these comparisons and is given by the formula:

$$D = \sqrt{d'S^{-1}d}$$

where $d$ is the vector of univariate standardised differences (Cohen's $d$) and $S$ is the correlation matrix.

We calculated the multivariate generalisation ($D$ measure) of personality differences in both samples, factoring in changes between the groups across all five factors of personality. When evaluated in this way, personality differences observed in both samples are considerably larger than some of the Cohen's $d$ effect sizes in isolation. The resulting multivariate effect sizes were calculated as $D = .69$ in the exploratory sample and $D = .23$ in the confirmatory sample. While significant differences were observed in levels of conscientiousness between the two groups, the overall differences in personality are

Table 3 **Results from a binary logistic model ($X^2(9, N = 617^{[a]}) = 20.51, p = .015$). This controls for a number of other variables that may also predict watch wearing.**

| Variables | $\beta$ | S.E. | Wald | Sig[*] | Exp ($\beta$) |
|---|---|---|---|---|---|
| **Gender** | .243 | .182 | 1.781 | .182 | 1.276 |
| **Age** | | | | | |
| 18–24 | | | 9.254 | **.026** | |
| 25–34 | −.348 | .221 | 2.479 | .115 | .706 |
| 35–49 | .184 | .204 | .818 | .366 | 1.203 |
| 55+ | .617 | .409 | 2.269 | .132 | 1.853 |
| **Personality** | | | | | |
| Extraversion | .000 | .056 | .000 | .999 | 1.000 |
| Agreeableness | .022 | .072 | .093 | .760 | 1.022 |
| Conscientiousness | .137 | .062 | 4.837 | **.028** | 1.147 |
| Emotional stability | .004 | .062 | .005 | .944 | 1.004 |
| Openness to experiences | −.113 | .076 | 2.210 | .137 | .893 |

Notes.
[a] $N = 617$ (22 participants from the original sample did not confirm their age and/or gender).
[*] Significant p-values are highlighted in bold.

not limited to a single personality factor. For example, in both samples watch wearers consistently produce lower extraversion and openness to experience scores.

## STUDY 3

The previous results lend strong support to the notion that people who choose to wear a watch also tend to rate themselves as more conscientious. While organisation is often considered as a lower-order facet score in many personality measures (e.g., as part of the HEXACO Personality Inventory; *Lee & Ashton, 2004*), higher levels of conscientiousness alone correlate with improved punctuality (*Back, Schmukle & Egloff, 2006*). *Ashton (1998)* also observed that conscientiousness was negatively associated with self-reported lateness in the workplace. Our final study therefore sought to investigate if punctuality is also related to watch wearing.

### Method
#### *Participants*

Ninety participants (29% male) who arrived to complete a separate experiment in the School of Psychology took part in this study. Their ages ranged from 17 to 48. All participants had previously visited the department on at least one previous occasion. This ensured that participant's were unlikely to become lost before an experiment was scheduled to start.

#### *Procedure*

Participants arriving at the School of Psychology for an unrelated experiment had their exact time of arrival recorded by the experimenter. Time of arrival was recorded as time-lag in minutes between the experiment appointment time and time of each participant's arrival. It was also noted whether they were a regular watch wearer.

#### *Results*

Participants who exceeded an early or late arrival time of $\pm 15$ min were removed from the analysis ($N = 5$) to ensure that data were normally distributed. On average, the remaining participants arrived 2.19 min before the appointed time ($SD = 5.95$). Mean punctuality scores (minutes late or early) were calculated for watch and non-watch wearers. A total of 34 watch wearers and 51 non-watch wearers arrival times were analysed (Fig. 1).

An independent sample $t$-test demonstrated a reliable difference in punctuality with participants in the watch-wearing group arriving significantly earlier ($M = 4.12$, $SD = 5.45$) in comparison to those who were not wearing a watch ($M = .90$, $SD = 5.96$), ($t(83) = 2.52, p = .01; d = .55$).

## GENERAL DISCUSSION

Choosing to wear a watch appears to act as a social marker for an individual who is likely to be more conscientious. A further replication across a larger sample supports this conclusion. We also observed consistent multivariate differences in personality between the two groups with watch wearers showing lower levels of extraversion and openness. Finally,

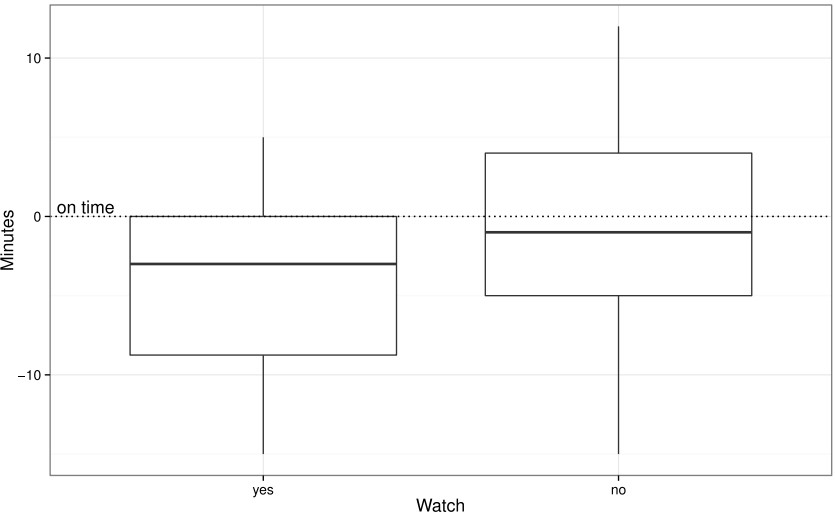

**Figure 1** **Differences in arrival times between watch and non-watch wearers.**

watch wearers behave in way that is consistent with higher levels of conscientiousness by arriving at an appointment earlier than non-watch wearers.

While personality has previously been linked to time perception (e.g., *Rammsayer, 1997*), this is the first study to link personality with the absence or presence of an everyday time cue. Higher levels of conscientiousness have previously been associated with increased levels of self-organisation in a variety of contexts and watch wearing may be an additional purchase decision that interacts with other related individual differences (*Aaker, 1997*). Conscientiousness alone is made up of many sub-facets of personality and one of these may play a more important role in watch wearing than others (e.g., organisation, diligence and perfectionism; *Lee & Ashton, 2004*).

These results could also be considered in the context of enclothed cognition, that is, the influence clothes or fashion accessories can have on a wearer's psychological processes. *Adam & Galinsky (2012)* propose that changes in cognition depend on both the symbolic meaning and physical experience of wearing different types of clothes, but this could also apply to wristwatches. As a fashion accessory, or expression of social status the act of wearing a watch may provide an additional, albeit implicit cognitive impact on wearers, which makes them more conscientious and better planners. In terms of punctuality specifically, appointment type may be an important factor to consider in future research, but these results are consistent with research demonstrating that personality is likely to be important when considering punctuality in isolation (*Back, Schmukle & Egloff, 2006*). Even if conscientious individuals are delayed, they will be dutiful enough to try to limit their lateness. In addition, our effect size relating to punctuality is far higher than previous correlations observed between conscientiousness and punctuality in a comparable sample by Back, Schmukle & Egloff and colleagues (*2006*).

The standard watch remains technologically simple, but this simplicity explains why countless manufactures of smartwatches are attempting to capitalize on this specific form

factor (*Fogg, 2009*). Such devices typically measure and provide additional feedback related to physical and physiological activity (e.g., heart rate). Interestingly, these devices are more likely to be purchased by those who already lead a healthy lifestyle (*Swan, 2009*). The desire to own or wear a standard wristwatch may therefore be driven by higher levels of conscientiousness in the first instance. Alternatively, the decision to purchase a watch may simply be motivated by a desire to know the time, become more organised and in turn attempt to become more conscientious.

Could the act of wearing a watch make an individual healthier or more conscientious? At present, this line of enquiry only extends to more simplistic devices like pedometers, where feedback correlates with an increase in physical activity, but not beyond the duration of the original intervention (*Bravata et al., 2007*). While watch wearing and smartwatch ownership correlate with increased levels of conscientiousness and health promoting behaviours, the direction of these relationships remains unclear, but worthy of further investigation. This is particularly relevant given existing links between the accuracy of clocks and long-term health outcomes (*Levine & Bartlett, 1984*; *Levine & Norenzayan, 1999*).

Another future direction for this research would be to explore the effect that watch wearing can have on first impressions and consider the relationship between self and others' perceptions of watch wearing. How such a time cue could influence other evaluative judgments by prompting attributions remains unclear. One might predict that the presence of a watch would serve to help improve an individual's first impression in a specific social context for example, at a job interview (*Chapplin et al., 2000*; *Dougherty, Turban & Callender, 1994*).

One limitation which could be levelled at this study is that some participants may own a mobile phone, but not a standard watch, which may act as a confounder because they still have rapid access to the time. However, 100% of our exploratory sample and 97.5% in our second sample also owned a mobile phone so this is unlikely to have been an influencing factor. It is worth noting, however, that the effect size relating to differences in conscientiousness reduced considerably between our exploratory and confirmatory samples. While the effect size is reduced in our larger sample, small effects could have larger aggregated consequences. For example, the short nature of the personality measure chosen suggests that a larger effect may be observed if a more in-depth measure of personality was deployed, but this may have limited our sample size. For now, we simply wanted to demonstrate that our exploratory findings could be replicated in a further independent sample using an identical measure of personality.

A second limitation concerns the reasons behind watch ownership. While an alternative explanation might conclude that choosing to wear a watch is related to social status and not a desire to know the time, this argument does not chime with the consistency of our results reported here. This is particularly pertinent when considered alongside our behavioural measure. However, we cannot rule this additional explanation out completely.

In sum, wearing a device that tells the time on the wrist is likely to remain an important tool for the foreseeable future and to our knowledge this is the first study to demonstrate a link between watch wearing, personality and related behaviour (*Anwar,*

*2012*). Specifically, watch wearers from a variety of backgrounds elicit significantly higher levels of conscientiousness and lower levels of extraversion and openness. They also arrive earlier for appointments. From the present data, it is not clear whether being conscientious inclines a person to wear a watch, or whether wearing a watch makes a person more conscientious. Whichever the direction of the relationship, watch wearing is a valid external marker of both personality and associated behaviour.

## ACKNOWLEDGEMENTS
We thank Sally Andrews and Lukasz Piwek for helpful comments on an earlier version of this manuscript.

### Funding
This research was supported by an ESRC studentship to David Ellis (ES/G017271/1), and an ESRC research grant to Rob Jenkins (RES-060-25-0010). The funders had no role in study design, data collection and analysis, decision to publish, or preparation of the manuscript.

### Grant Disclosures
The following grant information was disclosed by the authors:
ESRC: ES/G017271/1, RES-060-25-0010.

### Competing Interests
The authors declare there are no competing interests.

### Author Contributions
- David A. Ellis conceived and designed the experiments, performed the experiments, analyzed the data, contributed reagents/materials/analysis tools, wrote the paper, prepared figures and/or tables, reviewed drafts of the paper.
- Rob Jenkins conceived and designed the experiments, contributed reagents/materials/analysis tools, wrote the paper, reviewed drafts of the paper.

### Human Ethics
The following information was supplied relating to ethical approvals (i.e., approving body and any reference numbers):

The University of Glasgow, College of Science & Engineering Ethics Committee approved all research (2013-4641). Participants were informed about procedures in detail and provided written informed consent.

### Supplemental Information
Supplemental information for this article can be found online at http://dx.doi.org/10.7717/peerj.1210#supplemental-information.

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
