# Peer review of "Watch-wearing as a marker of conscientiousness"

_PeerJ, doi:10.7717/peerj.1210_

## Round 0.1 · original submission · Major Revisions

Reviewer 1 mentioned that there's only one variable to predict personality. Please either increase that, or report the precision/recall of prediction as suggested.

There were additional comments, left by Lukasz Piwek, on the preprint version of your article here https://peerj.com/preprints/989v1/#feedback and I suggest you also address them in a revised version of your manuscript. They are reproduced, for the record, below.

"This very interesting argument proposed by authors, that wearing wristwatches is associated with higher conscientiousness and punctuality, could be further elaborated in the context of ‘enclothed cognition’. Adam & Galinsky (2012) used term ‘enclothed cognition’ to describe the systematic influence that clothes have on the wearer’s psychological processes. For instance, Adam & Galinsky (2012) showed that wearing a lab coat described as a doctor’s coat increased sustained attention compared to wearing a lab coat described as a painter’s coat, and compared to simply seeing or even identifying with a lab coat described as a doctor’s coat. They argued that ‘enclothed cognition’ depends on both the symbolic meaning and the physical experience of wearing the clothes.

Its plausible to consider watches are a part of clothing - an accessory that may be a form of a fashion statement or an expression of social status (as pointed by authors in the Discussion). Therefore ‘enclothed cognition’ could add to a possibility that simply wearing a watch has some level of continuous, implicit, and cognitive impact on the wearers (like making people more conscientious, punctual or even healthier).

Going a (bit philosophical) step further, an idea that watch extends some subtle aspects of everyday psychological functioning is in line with Marshall McLuhan (1967) notion of the ‘extended mind’. McLuhan (1967) concept has been increasingly used to examine psychological impact of various technological solutions. For instance, in a recent study Barr et al. (2015) showed that those who think more intuitively and less analytically when given reasoning problems were more likely to rely on their Smartphones (i.e., extended mind) for information in their everyday lives. While the watch is a much simpler technological extension of mind, it works perfect as an instant access point to the current time. From the perspective of persuasive technology (Fogg, 2009), such simplicity behind watch design is probably the key to its effortless blending into the wearers everyday life. Which adds to the reason why a countless manufacturers of new-breed digital wearable smartwatches are trying to capitalise on this specific form factor.

References:

Adam, H., & Galinsky, A. D. (2012). Enclothed cognition. Journal of Experimental Social Psychology, 48(4), 918–925.

Barr, N., Pennycook, G., Stolz, J. a., & Fugelsang, J. a. (2015). The brain in your pocket: Evidence that Smartphones are used to supplant thinking. Computers in Human Behavior, 48, 473–480.

McLuhan, M., & Fiore, Q. (1967). The Medium is the Massage. London: Penguin Group.

Fogg, B. J. (2009). A behavior model for persuasive design. In Proceedings of the 4th International Conference on Persuasive Technology - Persuasive ’09. New York: ACM Press.

Reviewer 1 ·

Basic reporting

No Comments

Experimental design

It is quite possible that wearing a traditional wristwatch is just for show-off. For the experimental design, it is very important to make sure to eliminate irrelavant factors, or put other variables into consideration. Since all data are acquired by self-report, the authors should avoid common method variance.

Validity of the findings

Only correlation results are reported in this paper, and it is inappropriate to use the only variable (wearing a traditional wristwatch or not) to predict personality. To do so, the precision/recall of prediction should be reported to support the finding.

Additional comments

No Comments.

Reviewer 2 ·

Basic reporting

No Comments.

Experimental design

No Comments.

Validity of the findings

No Comments.

Additional comments

This study is meaningful, though there are a few points that I am not sure when reading the manuscript.
1. What do you mean by the note "The TIPI was intentionally designed to produce low coefficient alphas, which are themselves misleading when calculated on scales with a reduced number of items"? How about the coefficient alphas in this study?(see line 117-118)

2.In study 1 and 2, the wearing of wristwatch could predict whether the subject is more conscientious or not. However, is there any important factor such as profession or vocation of the participants playing a part or/and interacting? Yet you haven't spell out the information of the participants in Study2, and only mentioned "students studying psychology" in Study 1. What biases were you aware of in your sample?

3.In the discussion, you mentioned “ while traditional watch wearing and smart watch ownership correlate with increased levels of conscientiousness and health promoting behaviors, the direction of these relationships remains unclear”(line 234-236). Since the direction of the relationship between watch wearing and conscientiousness is unclear, the word ‘ predict’ in the title maybe not accurate.

4. In the multivariate analysis of study 2, you need to describe more clearly to the readers the process of calculating the multivariate generalisation, especially the D measures in two samples.

---

## Round 0.2 · accepted · Accept

Thanks for addressing the points raised by the reviewers.

Reviewer 1 ·

Basic reporting

This paper is well organized and written.

Experimental design

The research problem is clearly defined. The authors have added more variables into the analysis, and the results sound reasonable.

Validity of the findings

The conclusion made from the data is reasonable.

Additional comments

This paper investigates an interesting topic, the experiment has been well designed, and the data has been processed correctly.